# A Combinatorial Algorithm for Approximating the Optimal Transport in the Parallel and MPC Settings

**Nathaniel Lahn**
Radford University
nlahn@radford.edu

**Sharath Raghvendra**
Virginia Tech
sharathr@vt.edu

**Kaiyi Zhang**[*]
Virginia Tech
kaiyiz@vt.edu

## Abstract

Optimal Transport is a popular distance metric for measuring similarity between distributions. Exact and approximate combinatorial algorithms for computing the optimal transport distance are hard to parallelize. This has motivated the development of numerical solvers (e.g. Sinkhorn method) that can exploit GPU parallelism and produce approximate solutions.

We introduce the first parallel combinatorial algorithm to find an additive $\varepsilon$-approximation of the OT distance. The parallel complexity of our algorithm is $O(\log(n)/\varepsilon^2)$ where $n$ is the total support size for the input distributions. In Massive Parallel Computation (MPC) frameworks such as Hadoop and MapReduce, our algorithm computes an $\varepsilon$-approximate transport plan in $O(\log(\log(n/\varepsilon))/\varepsilon^2)$ rounds with $O(n/\varepsilon)$ space per machine; all prior algorithms in the MPC framework take $\Omega(\log n)$ rounds. We also provide a GPU-friendly matrix-based interpretation of our algorithm where each step of the algorithm is row or column manipulation of the matrix. Experiments suggest that our combinatorial algorithm is faster than the state-of-the-art approximate solvers in the GPU, especially for higher values of $n$.

## 1 Introduction

Optimal transport (OT) is a useful metric for measuring similarity between distributions and has numerous applications [2, 4, 5, 9, 12, 29], including image retrieval [28], GAN training [23], and interpolation between distributions [7]. Given two distributions $\mu$ and $\nu$, this metric captures the minimum-cost plan for transporting mass from $\mu$ to $\nu$.

More formally, in the *optimal transport* problem, we are given two discrete distributions $\mu$ and $\nu$ whose supports are the point sets $A$ and $B$, respectively. For each point $a \in A$ (resp. $b \in B$), we associate a probability of $\mu_a$ (resp. $\nu_b$) with it such that $\sum_{a \in A} \mu_a = \sum_{b \in B} \nu_b = 1$. We refer to each point of $A$ as a demand point and each point in $B$ as a supply point. For any edge $(a, b) \in A \times B$, we are given a cost $c(a, b)$; we assume that the costs are scaled so that the largest cost edge is $1$. Let $\beta c(a, b)$ be the cost of transporting a supply amount of $\beta$ from $b$ to $a$. A transport plan is a function $\sigma : A \times B \to \mathbb{R}_{\geq 0}$ that assigns a non-negative value to each edge of $G$, indicating the amount of supply transported along the edge. The transport plan $\sigma$ is such that the total supplies transported into (resp. from) any demand (resp. supply) node $a \in A$ (resp. $b \in B$) is bounded by the demand (resp. supply) at $a$ (resp. $b$). The cost of the transport plan, denoted by $c(\sigma)$, is given by $\sum_{(a,b) \in A \times B} \sigma(a, b) c(a, b)$. In this optimal transport problem, we are interested in finding a minimum-cost transport plan that transports all of the supply, denoted by $\sigma^*$. We also define an

---

[*]Following convention from Theoretical Computer Science, all authors are ordered in alphabetical order.
The code used for the experiments reported in this paper is available at: https://github.com/kaiyiz/Combinatorial-Parallel-OT.

$\varepsilon$-approximate transport plan to be any transport plan $\sigma$ with a cost $c(\sigma) \le c(\sigma^*) + \varepsilon$ that transports all of the supply.

The special case where $A$ and $B$ each contain $n$ points each and where every point in $A$ (resp. $B$) has a demand of $1/n$ (resp. supply of $1/n$) is called the *assignment problem*. In this special case, there is an optimal transport plan with a special structure; specifically, when there are $n$ vertex-disjoint edges $(a, b)$ with $\sigma(a, b) = 1/n$, they form a *perfect matching*. Let the cost of any matching $M$, denoted by $c(M)$ be the total cost of all of its edges, i.e.,

$$c(M) = \sum_{(a,b) \in M} c(a, b).$$

Given a perfect matching $M$, the cost of the corresponding transport plan is simply $1/n \sum_{(a,b) \in M} c(a, b) = (1/n)c(M)$. For simplicity in exposition, in the context of the assignment problem, we will uniformly scale all the demands and supplies from $1/n$ to $1$. This does not change the optimal transport plan. It, however, increases the cost of the optimal transport plan to $c(M)$, an increase by a factor of $n$. Thus, for the assignment problem, finding the optimal transport plan is equivalent to finding a minimum-cost perfect matching $M^*$.

Similarly, for an $\varepsilon > 0$, after scaling the demands and supplies from $1/n$ to $1$, an $\varepsilon$-approximate transport plan corresponds to a perfect matching $M$ with cost $c(M) \le c(M^*) + \varepsilon n$. We refer to such a perfect matching as $\varepsilon$-*approximate matching*. Thus, for the assignment problem, finding an $\varepsilon$-approximate transport plan corresponds to finding an $\varepsilon$-approximate matching.

**Related Work:** For discrete distributions, the optimal transport problem can be formulated as a minimum-cost flow problem and solved using any LP-solver. The best-known exact and approximate solver for optimal transport, as well as the assignment problem, are graph-based combinatorial algorithms [25, 19, 17, 27]. These solvers, however, are known to be difficult to parallelize. For instance, the best $\varepsilon$-approximate OT solver, in terms of sequential running time, was given by Lahn *et al.* [19]. This combinatorial algorithm runs in $O(n^2/\varepsilon + n/\varepsilon^2)$ time, and is a non-trivial adaptation of the classical combinatorial exact algorithm by Gabow and Tarjan (GT-algorithm) for the transportation problem [13]. The Lahn *et al.* algorithm runs no more than $\lfloor 2/\varepsilon \rfloor + 1$ iterations, where each iteration executes a Dijkstra's shortest path search to find and augment along a set of "augmenting paths". This algorithm is the state-of-the-art in the sequential setting; see [22] for a discussion on the various algorithms. Unfortunately, however, the flow augmentations have to be done in a sequential manner, making this algorithm hard to parallelize.

Motivated by the need for efficient scalable solutions, machine learning researchers have designed highly parallelizable approximate solvers that generate an $\varepsilon$-approximate transport plan in $\tilde{O}(n^2/\varepsilon^{O(1)})$ sequential time and $\tilde{O}(1/\varepsilon^{O(1)})$ parallel time. Perhaps the most successful among these is an entropy regularized version of the optimal transport, which can be solved using the Sinkhorn-Knopp method [1, 8] and produces an $\varepsilon$-approximation to the optimal transport in $\tilde{O}(n^2/\varepsilon^2)$ sequential time and $\tilde{O}(1/\varepsilon^2)$ parallel time [10]. The simplicity of this algorithm has lead to an efficient GPU implementation. From a theoretical standpoint, Jambulapati *et al.* [16] designed a dual extrapolation algorithm using an area convex mapping [31] to achieve an improved parallel complexity of $O((\log(n) \log \log n)/\varepsilon)$. However, as noted by Lin *et al.* [22], despite its sound theoretical guarantees, the lack of simplicity and the difficulties of implementation make this algorithm by Jambulapati *et al.* less competitive, and the Sinkhorn algorithm remains the state-of-the-art for approximating the Optimal Transport on GPUs.

Despite being the state-of-the-art in sequential settings, combinatorial algorithms have remained difficult to parallelize and all known exact or approximate combinatorial algorithms run in only slightly sub-linear parallel time [14]. In this paper, we design the first parallel combinatorial algorithm that takes only $O(\log(n)/\varepsilon^2)$ parallel time and finds an $\varepsilon$-approximation transport plan.

Our algorithm also improves upon existing algorithms in the massive parallel computation frameworks such as Hadoop, MapReduce, Dryad and Spark. In the Massively Parallel Computing (MPC) model, we are given a set of machines, where each machine has a bounded amount of memory. For this model, the base assumption is that communication between machines is the performance bottleneck, and the goal is to minimize the number of synchronized *communication rounds*, where each round consists of a period of local computation on each machine, followed by a period of message communication between machines. It is well known that any standard parallel algorithm that takes $O(f(n))$ time can

be directly translated to an algorithm under the MPC model that runs in $O(f(n))$ communication rounds. However, algorithms that are more specialized for the MPC model can acheive drastically faster computation times, often having a sub logarithmic number of rounds. For example, it has long been known how to compute a maximal matching in $O(\log n)$ parallel time [15], but only recently was a breakthrough made that shows how to compute a maximal matching in $O(\log\log n)$ rounds under the MPC model [3]. Maximal matching is a substantially simpler problem than both the assignment and OT problems. For these OT problem, known parallel $\varepsilon$-approximation algorithms immediately yield an $\tilde{O}(\log(n)/\varepsilon)$ round algorithm for the MPC model [16]. To our knowledge, no specialized MPC algorithms are known for the either problem. Thus, we provide the first sub-logarithmic round $\varepsilon$-approximation algorithm for both the assignment and OT problems. We obtain this bound by leveraging the recent breakthrough MPC algorithm for maximal matching by [3].

**Our Results:** In this paper, we present a very simple combinatorial algorithm to compute an $\varepsilon$-approximate transport plan in $O(n^2/\varepsilon^2)$ sequential time and $O(\log(n)/\varepsilon^2)$ parallel time. For the special case of the assignment problem, the sequential execution time of our algorithm improves to $O(n^2/\varepsilon)$. We also provide a GPU implementation of our algorithm that outperforms the implementation of the Sinkhorn algorithm provided by Python Optimal Transport library [11].

Our algorithm also extends to the well-known Massive Parallel Computation (MPC) frameworks such as MapReduce, Hadoop, Dryad and Spark. In the MPC model, our algorithm computes a $\varepsilon$-approximate transport plan in $O(\log(\log n)/\varepsilon^2)$ rounds with $O(n)$ memory per machine. Our algorithm is based on the popular push-relabel framework [14] for computing minimum-cost flow.

**Theorem 1.1.** *Given an $\varepsilon > 0$, there is an algorithm that computes an $\varepsilon$-approximate matching in $O(n^2/\varepsilon)$ time. Furthermore, one can execute this algorithm in expected $O(\log(n)/\varepsilon^2)$ parallel time or in expected $O(\log(\log n)/\varepsilon^2)$ rounds, with $O(n)$ memory per machine, in the MPC model.*

*Extension to the Optimal Transport problem:* For any $\varepsilon > 0$, Lahn *et al.* [19] showed that computing an $\varepsilon$-approximation of the optimal transport between two discrete distributions containing $n$ points in their support reduces to an instance of an unbalanced assignment problem with $n/\varepsilon$ points. We apply this reduction and slightly adapt our algorithm from Theorem 1.1 to obtain our result for the optimal transport problem (Theorem 1.2). In this paper, we present details of our algorithm for the assignment problem. Details of the adaptation of our algorithm to the optimal transport by using the reduction of [19] is presented in the Section B of the appendix.

**Theorem 1.2.** *Given an $\varepsilon > 0$, there is an algorithm that computes an $\varepsilon$-approximate transport plan in $O(n^2/\varepsilon^2)$ time. Furthermore, one can execute this algorithm in expected $O(\log(n)/\varepsilon^2)$ parallel time or in $O(\log(\log(n/\varepsilon))/\varepsilon^2)$ rounds with $O(n/\varepsilon)$ memory per machine in the MPC model.*

From a theoretical stand-point, we provide the first parallel combinatorial algorithm for approximating the optimal transport with a expected parallel execution time of $O(\log(n)/\varepsilon^2)$. The sequential execution time of $O(n^2/\varepsilon)$ for our algorithm for the assignment problem matches with the current state-of-the-art for the problem [16, 19]. We also provide the first sub-logarithmic round algorithm that approximates the optimal transport plan in the MPC model.

From a practical stand-point, for both the assignment problem and the OT problem, we provide an implementation that exploits GPU parallelism. Experiments suggest that both of our GPU implementations outperform the GPU implementation of the state-of-the-art Sinkhorn algorithm provided by the Python Optimal Transport library [11] in terms of running time, while achieving the same level of accuracy.

**Our Approach:** Our algorithmic approach is based on the popular push-relabel framework for computing network flows. For the assignment problem, our algorithm maintains a matching $M$ and a set of dual weights $y(\cdot)$ on vertices of $A \cup B$. The algorithm runs in $O(1/\varepsilon^2)$ iterations and in each iteration, it executes three steps. First, it greedily computes a maximal matching $M'$. In the second step, it uses $M'$ to update the matching $M$ (the push step). Finally, it updates the dual weights (relabel step). Our proof of correctness is based on the standard dual feasibility conditions used to compute minimum-cost maximum cardinality matchings, with some modifications made to better accommodate our additive-approximate setting. Our main technical difference from standard push-relabel techniques is the novel running time analysis for the additive approximate setting. In particular, we show that the number of iterations required by our algorithm is just $O(1/\varepsilon^2)$. Within each iteration, the push and relabel steps take only $O(n)$ sequential time and $O(1)$ parallel time. The

only non-trivial step, therefore, is the computation of a maximal matching which can be done in $O(n^2)$ sequential time and $O(\log n)$ parallel time [15]. Maximal matchings can also be computed in $O(\log \log n)$ rounds in the massively parallel computation (MPC) model [3]. As a result, our algorithm can also be executed in $O(\log(\log n)/\varepsilon^2)$ rounds in the MPC model, for the assignment problem. We extend our algorithm to also approximate the optimal transport plan by using the reduction of Lahn *et al.* [19] (see Section B of the appendix for details).

**Organization:** In Section 2.1, we present the definitions required to describe our algorithm. In Section 2.2, we present our algorithm for the assignment problem. For simplicity of exposition, we present an algorithm that computes a $3\varepsilon$-approximation of the optimal solution to the assignment problem. To obtain an $\varepsilon$-approximation, one can simply choose the error factor in the algorithm to be $\varepsilon/3$. In Section 3, we prove the sequential complexity of our algorithm for the assignment problem. In Section 4, we analyze the complexity of our algorithm in the parallel and MPC settings and also describe a GPU-friendly implementation of our matching algorithm. Finally, we present the experimental results in Section 5. In the appendix, Section B, we extend our algorithm to the optimal transport problem. All missing proofs can also be found in the appendix, Section A.

## 2 Algorithm

In this section, given an input to the assignment problem and a value $0 < \varepsilon < 1$, we present an algorithm that computes a $3\varepsilon$-approximate matching.

### 2.1 Preliminaries

We begin by introducing the terminologies required to understand our algorithm for the assignment problem. For any matching $M$, we say that any vertex $v \in A \cup B$ is *free* if $v$ is not matched in $M$ and *matched* otherwise. Our algorithm critically uses the notion of a maximal matching which we introduce next. For any bipartite graph that is not necessarily complete, any matching $M$ is *maximal* if and only if at least one end point of every edge in the graph is matched in $M$. Thus, if a matching is not maximal, there is at least one edge between two free vertices. One can, therefore, compute a maximal matching in a greedy fashion by iteratively picking such an edge and adding it to the matching.

For every edge $(u, v) \in A \times B$, we transform its cost so that it becomes an integer multiple of $\varepsilon$ as follows:

$$\overline{c}(u, v) = \varepsilon \lfloor c(u, v)/\varepsilon \rfloor \tag{1}$$

The rounding of edge costs may introduce an error that is bounded by $\varepsilon$ for each edge and by at most $\varepsilon n$ for any matching. Our algorithm assigns a dual weight $y(v)$ for every $v \in A \cup B$ such that a set of relaxed dual feasibility conditions are satisfied. A matching $M$ along with dual weights $y(\cdot)$ is $\varepsilon$-feasible if, for every edge $(a, b) \in A \times B$,

$$y(a) + y(b) \leq \overline{c}(a, b) + \varepsilon \qquad \text{if } (a, b) \notin M \tag{2}$$
$$y(a) + y(b) = \overline{c}(a, b) \qquad \text{if } (a, b) \in M \tag{3}$$

In Lemma 3.1, we show that any $\varepsilon$-feasible matching produced by our algorithm has a cost within an additive error of $\varepsilon$ from the optimal solution with respect to the costs $\overline{c}(\cdot, \cdot)$. For any edge $(u, v)$, we define its *slack* $s(u, v)$ to be $0$ if $(u, v) \in M$. Otherwise, if $(u, v) \notin M$, we set its slack to be $s(u, v) = \overline{c}(u, v) + \varepsilon - y(u) - y(v)$. We say that $(u, v)$ is admissible if the slack on the edge is $0$.

We observe that any matching $M$ whose cardinality is at least $(1 - \varepsilon)n$ can be converted into a perfect matching simply by arbitrarily matching the remaining $\varepsilon n$ free vertices. The cost of any edge is at most $1$, and so, this increases the cost of the matching $M$ by at most $\varepsilon n$. In addition to this, the rounding of costs from $c(\cdot, \cdot)$ to $\overline{c}(\cdot, \cdot)$ also introduces an increase of cost by $\varepsilon n$. Finally, the $\varepsilon$-feasibility conditions introduced an additional additive error of $\varepsilon n$, for a total error of $3\varepsilon n$, as desired. Thus, in the rest of this section, we present an algorithm that computes an $\varepsilon$-feasible matching of cardinality at least $(1 - \varepsilon)n$, which has a cost no more than $\varepsilon n$ above the optimal matching's cost with respect to $\overline{c}(\cdot, \cdot)$.

## 2.2 Algorithm Details

Initially, we set the dual weight of every vertex $b \in B$ to be $\varepsilon$ and every vertex $a \in A$ to be 0. We initialize $M$ to $\emptyset$. Our initial choice of $M$ and the dual weights satisfies (2) and (3). Our algorithm executes iterations, which we will call *phases*. Within each phase, the algorithm constructs the set $B'$, which consists of all free vertices of $B$. If $|B'| \leq \varepsilon n$, then $M$ is an $\varepsilon$-feasible matching of cardinality at least $(1 - \varepsilon)n$, and the algorithm will arbitrarily match the remaining free vertices and return the resulting matching. Otherwise, the algorithm computes the subset $E' \subseteq E$ of admissible edges with at least one end point in $B'$. Let $A' = \{a \mid a \in A \text{ and } (a, b) \in E'\}$, i.e., the set of points of $A$ that participate in at least one edge in $E'$. For each phase, the algorithm executes the following steps:

(I) *Greedy step:* Computes a maximal (i.e., greedy) matching $M'$ in the graph $G'(A' \cup B', E')$.

(II) *Matching Update:* Let $A''$ be the set of points of $A'$ that are matched in both $M$ and $M'$ and let $M''$ be the edges of $M$ that are incident on some vertex of $A''$. The algorithm adds the edges of $M'$ to $M$ and deletes the edges of $M''$ from $M$.

(III) *Dual Update:*

    a. For every edge $(a, b) \in M'$, the algorithm sets $y(a) \leftarrow y(a) - \varepsilon$, and

    b. For every vertex $b \in B'$ that is free with respect to $M'$, the algorithm sets $y(b) \leftarrow y(b) + \varepsilon$.

In each phase, the matching update step will add edges of $M'$ to $M$ and remove edges of $M''$ from $M$. By construction, the updated set $M$ is a matching. Furthermore, every vertex of $A$ that was matched prior to the update continues to be matched after the update.

**Lemma 2.1.** *The new set $M$ of edges obtained after Step (II) is a matching. Furthermore, any vertex of $A$ that was matched prior to Step (II) will continue to be matched after the execution of Step (II).*

The dual update step increases or reduces dual weights by $\varepsilon$. Therefore, the dual weights always remain an integer multiple of $\varepsilon$.

The algorithm maintains the following invariants:

(I1) The dual weight of every vertex in $B$ (resp. $A$) is non-negative (resp. non-positive). Furthermore, every free vertex of $A$ has a dual weight of 0.

(I2) The matching $M$ and a set of dual weights $y(\cdot)$ is $\varepsilon$-feasible.

Proofs of these invariants can be found in the appendix Section A.1.

# 3 Analysis

Next, in Section 3.1, we use invariants (I1) and (I2) to show that the algorithm produces a matching with the desired accuracy. In Section 3.2, we use the invariants to bound the sequential and parallel execution times of our algorithm.

## 3.1 Accuracy

As stated in Section 2.1, the rounding of costs from $c(\cdot, \cdot)$ to $\overline{c}(\cdot, \cdot)$ introduces an error of $\varepsilon n$. Furthermore, after obtaining a matching of size at least $(1 - \varepsilon)n$, the cost of arbitrarily matching the last $\varepsilon n$ vertices is no more than $\varepsilon n$. From the following lemma, we can conclude that the total error in the matching computed by our algorithm is no more than $+3\varepsilon n$. Proof of this Lemma can be found in the appendix Section A.2

**Lemma 3.1.** *The $\varepsilon$-feasible matching of size at least $(1 - \varepsilon)n$ that is produced by the main routine of our algorithm is within an additive error of $\varepsilon n$ from the optimal matching with respect to the rounded costs $\overline{c}(\cdot, \cdot)$*

## 3.2 Efficiency

Suppose there $t$ phases executed by the algorithm. We use $n_i$ to denote the size of $B'$ in phase $i$. By the termination condition, each phase is executed only if $B'$ has more than $\varepsilon n$ vertices, i.e.,

$n_i > \varepsilon n$. First, in Lemma 3.2 (appendix A.3), we show that the magnitude of the dual weight of any vertex cannot exceed $(1 + 2\varepsilon)$. This means the total dual weight magnitude over all vertices is upper bounded by $n(1 + 2\varepsilon)$. Furthermore, in Lemma 3.3 (appendix A.4) we show that, during phase $i$, the total dual weight magnitude increases by at least $\varepsilon n_i$. From this, we can conclude that

$$\sum_{i=1}^{t} n_i \leq n(1 + 2\varepsilon)/\varepsilon = O(n/\varepsilon). \tag{4}$$

Note that, since each $n_i \geq \varepsilon n$, we immediately get $t\varepsilon n \leq n(1+2\varepsilon)/\varepsilon$, or $t \leq (1+2\varepsilon)/\varepsilon^2 = O(1/\varepsilon^2)$. In order to get the total sequential execution time, we show, in Lemma 3.4 (appendix A.5) that each phase can be efficiently executed in $O(n \times n_i)$ time. Combining this with equation (4) gives an overall sequential execution time of $O(n(\sum_{i=1}^{t} n_i)) = O(n^2/\varepsilon)$.

**Lemma 3.2.** *For any vertex $v \in A \cup B$, the magnitude of its dual weight cannot exceed $1 + 2\varepsilon$, i.e., $|y(v)| \leq (1 + 2\varepsilon)$.*

**Lemma 3.3.** *The sum of the magnitude of the dual weights increases by at least $\varepsilon n_i$ in each iteration.*

**Lemma 3.4.** *The execution time of each phase is $O(n \times n_i)$ time.*

### 3.3 Analysis for the Unbalanced Case

In this section, we describe how the analysis of our matching algorithm can be extended to work for the unbalanced case, where $|A| \neq |B|$. This analysis is critical for proving the correctness of our optimal transport version of the algorithm. Without loss of generality, assume $|B| \leq |A| = n$. The overall description of the algorithm remains the same, except for the main routine of our algorithm produces an $\varepsilon$-feasible matching of size at least $(1 - \varepsilon)|B|$. The asymptotic running time of both the parallel and sequential algorithms remains unchanged. In the following lemma, we bound the additive error of our algorithm for the unbalanced case; the argument is very similar to Lemma 3.1.

**Lemma 3.5.** *Given an unbalanced input to the assignment problem with $|B| \leq |A|$, the $\varepsilon$-feasible matching of cardinality at least $(1 - \varepsilon)|B|$ that is returned by our algorithm is within an additive error of $\varepsilon|B|$ from the optimal matching with respect to the cost function $\overline{c}(\cdot, \cdot)$*

## 4 Parallel Algorithm

In this section, we describe how to parallelize the matching algorithm of Section 2.2 leading to the result of Theorem 1.1. Recall that each phase of this algorithm has three steps : (I) Greedy step, (II) Matching update, and (III) Dual update. Steps (II) and (III) are easily parallelizable using $O(1)$ time. However, step (I) is nontrivial to parallelize. Fortunately, Isreali and Itai gave a $O(\log n)$ randomized parallel algorithm for computing a maximal matching on an arbitrary graph [15]. Therefore, we can complete step (I) by applying their algorithm as a black box. However their algorithm is very generic, applying even for non-bipartite graphs. In Section 4.1, we use the Israeli Itai algorithm to bound the parallel complexity of our algorithm. In Section 4.2, we further parallelize the phases of our algorithm leading to a simplified variation of the Israeli Itai algorithm that is more suited for a practical implementation. Finally, in Section 4.3, we provide a matrix-based interpretation of our simplified algorithm, allowing the algorithm to be easily implemented for GPUs.

### 4.1 Analysis of our Algorithm for Parallel and MPC Models

The Israeli Itai algorithm is designed to work on an arbitrary graph $G(V, E)$, which may not be bipartite. It computes a maximal matching on $G$ in $O(\log n)$ iterations. By directly using their $O(\log n)$ algorithm for step (I) of our algorithm, we obtain an $O(\log n)$ parallel running time for each phase. Since our algorithm executes $O(1/\varepsilon^2)$ phases, we obtain a worst-case theoretical bound of $O(\log n/\varepsilon^2)$ for our algorithm.

We would also like to note that specialized algorithms for maximal matching exist for the MPC model as well. For example, it is possible to compute a maximal matching under the MPC model using just $O(\log(\log(\Delta)))$ rounds and $O(n)$ space per machine, where $\Delta$ is the maximum degree of the graph. [3]). As a result, we are able to achieve an algorithm in the MPC model that requires only $O(\log(\log n)/\varepsilon^2)$ rounds with $O(n)$ space per machine.

## 4.2 Simplifying the Parallel Implementation of our Algorithm

Instead of using Israeli Itai algorithm (which works for any arbitrary graph) as a black-box, we use a simpler adaptation of their algorithm for bipartite graphs. The Israeli Itai algorithm executes $O(\log n)$ iterations, where each iteration executes the following steps, using $O(1)$ parallel time:

  (i) Each vertex $u \in V$ selects an incident edge $(u, v)$ at random, and directs it from $u$ to $v$, yielding a directed subgraph $R \subseteq E$.

  (ii) Each vertex $u \in V$ selects, at random, one incoming edge from $R$. Let $S$ be the set of edges chosen by this step, with all directions removed. The graph $S$ has a maximum vertex degree of 2.

  (iii) Each vertex selects an edge of $S$ at random. Any edge that is selected by both of its endpoints is added to $M$, and any vertex matched by this step is removed for the next phase.

The Israeli Itai algorithm is designed to work for any graph that is not necessarily bipartite. In this situation, there is no way to partition the vertices into sets $A$ and $B$, and so it is necessary to consider edges as having two directions; a vertex $u$ could 'propose' an outgoing edge $(u, v)$ (see step (i)) and also 'receive' multiple proposals as incoming edges (see step (ii)). In the non-bipartite case, it is necessary for *every* vertex to both send and receive proposals; all vertices must be handled using a symmetric process. This results in a subgraph $S$, where each vertex could have a degree of 2, and an additional step is required to eliminate some of these edges in order to form a matching (see step (iii)).

However, in our situation, we are solely working with bipartite graphs. In this situation, the vertices are divided into two sets $A$ and $B$, and, as a result, we do not need to process each vertex in a symmetric fashion. Instead, we can allow one side to make proposals and the other side to receive proposals. As a result, for each iteration of the maximal matching algorithm, we can execute the following steps, which correspond to steps (i) and (ii) in the Israeli Itai algorithm.

  (a) Each vertex of $B$ selects, at random, an incident edge, yielding a subgraph $S$.

  (b) Each vertex of $A$, with degree at least one in $S$, arbitrarily selects an edge from $S$ and adds it to the matching.

Note that, after step (b) in our approach, each vertex has at most one incident edge selected. This alleviates the need for step (iii), since steps (a) and (b) alone immediately result in a matching.

In addition to our simplified approach to each iteration of the Israeli Itai algorithm, we make a second optimization: Within our algorithm, instead of waiting for the Israeli Itai algorithm to complete, which could require $O(\log n)$ iterations, our implementation *immediately* updates the matching and dual weights after each iteration before moving to the next iteration of the Israeli Itai algorithm. While this increases the number of phases, each phase becomes very simple, taking only $O(1)$ time, resulting in practical improvements. We believe that this additional source of parallelization could lead to asymptotic improvements to the parallel complexity of our algorithm. However, the proofs of Israeli Itai do not readily extend to this modified algorithm. Obtaining a tight bound on the parallel complexity of our modified algorithm is an important open question.

## 4.3 A GPU-Friendly Implementation

Thus far, we have described our algorithm using graph theoretic notations. However, in practice, it is important for our algorithm to have an efficient GPU-based implementation. In this section, we provide a matrix-based implementation of our algorithm, using the simplified maximal matching approach described in Section 4.2.

In Algorithm 1, we provide a pseudocode of our simplified algorithm. The algorithm resembles the one described in Section 2.2, except for the differences described in Section 4.2. It assumes that the costs were already rounded to each be an even multiple of $\varepsilon$. The matching returned by the algorithm has cardinality at least $(1 - \varepsilon)n$ and a cost at most $\varepsilon n$ above the optimal cost. The algorithm, as written, does not maintain dual weights explicitly, but if dual weights are required as part of the output, then, as discussed below, the algorithm can be modified to keep track of them. Note that all operations in this psuedocode are based on relatively simple matrix operations, and can be implemented easily on a GPU.

---

**Algorithm 1** Approximate Bipartite Matching

---

1: Input: $W \in \mathbb{R}^{n \times n}_{\geq 0}, \delta \in \mathbb{R}_{>0}$
2: $M \leftarrow \mathbf{0}_{n \times n}, S \leftarrow W$
3: **while** $M$ has more than $\varepsilon n$ columns with all 0's **do**
4:     $P \leftarrow \mathbf{0}_{n \times n}$
5:     **for all** columns $b$ with all zero entries in $M$ **do**
6:         Randomly select a row $a$ such that $S_{a,b} = 0$
7:         $P_{a,b} \leftarrow 1$
8:     **for all** $a, b \in [1..n]$ **do**
9:         $M_{a,b} \leftarrow 1$ **if** $M_{a,b} = 1$ **and** row $a$ of $P$ has all 0's
10:        **otherwise** $M_{a,b} \leftarrow P_{a,b}$
11:    **for all** rows $a$ in $P$ with at least one 1 **do**
12:        Add $\varepsilon$ to every entry in row $a$ of $S$
13:    **for all** columns $b$ in $M$ with all 0 entries **do**
14:        $\rho \leftarrow$ the minimum entry in column $b$ of $S$
15:        Decrease every entry in column $b$ of $S$ by $\rho$
16: **return** $M$

---

The algorithm takes as input an $n \times n$ cost matrix $W$ and a value for the error parameter $\delta$, and returns an $n \times n$ bit matrix that describes the matching, where an edge $(i, j)$ is in the matching if and only if the value at row $i$ and column $j$ is a 1. We use $\mathbf{1}_n$ to represent a row vector containing all 1's, and $\mathbf{1}_{m \times n}$ to represent an $m$ by $n$ matrix of all 1's. We use similar notation for vectors and matrices of all 0's. When considering an $n \times n$ matrix, we follow a convention that each row corresponds to a vertex of $A$, and each column corresponds to a vertex of $B$.

Next, we explain further each part of the algorithm. Line 3 initializes the slack matrix $S$ to reflect the initial slacks of all edges, which are initially non-matching. Throughout the algorithm, the matrix $S$ will reflect the slack with respect to the edge 'as if it were a non-matching edge', i.e., $S_{i,j} = W_{i,j} + \varepsilon - y(a) - y(b)$, regardless of the matching status of the edge. This makes the slacks easier to track without the need to explicitly maintain dual weights.

The main loop of the algorithm, beginning at line 4, specifies the stopping condition of the algorithm. The algorithm terminates once at most $\varepsilon n$ free vertices of $B$ remain. Each iteration of this main loop is a *phase*. Lines 5–7 compute a matrix $P$, which represents a set of edges that will be added to $M$ during the current phase. This edge set will be a matching on the admissible edges that are incident on free vertices of $B$. This corresponds to Step (I) of the algorithm from Section 2.2, except for $P$ is not necessarily maximal. Lines 8–10 update the matching $M$ by adding edges of $P$ to $M$, and removing any preexisting edges of $M$ that are matched in $P$. This corresponds to Step (2) of the algorithm from Section 2.2. Finally, lines 11–15 update the slacks to reflect dual weight adjustments, corresponding to Step (3) of the algorithm from Section 2.2. However, instead of tracking the dual weights explicitly, we simply update the slacks directly. Note that, when updating the slacks on edges incident on free vertices of $B$, we include a slight change to Step (3). Instead of increasing the dual weight of free vertices of $B$ by *exactly* $\varepsilon$, we increase it as much as possible. For some free vertices of $B$, the increase will be 0 (since $P$ is not maximal), but it is also possible, in practice, for the increase to be larger than $\varepsilon$.

## 5 Experiments

In this section, we present our experimental results. We implemented the parallel version of our algorithm for both the assignment problem as well as the OT problem. Both implementations are written in Python using the PyTorch library, which supports GPU operations. We compare these implementations of our algorithm to the Sinkhorn algorithm implementation in the Python Optimal Transport (POT) library [11]. This Sinkhorn implementation also uses PyTorch. Additionally, we compare our algorithm to a CUDA based GPU implementation and present the results of this comparison in appendix section C.3.

Our experiments are run with an Intel Xeon E5-2680v4 2.4GHz chip and an Nvidia Tesla V100 GPU. We ran our algorithms using both real and synthetic data, including four different settings:

an assignment problem between randomly generated synthetic point sets, an OT problem between randomly generated point sets, each having randomly assigned demands and supplies, an assignment problem formed from two sets of MNIST images, and an OT problem between two text word embeddings. For each setup, we generated input data and computed the assignment or OT cost using our algorithm with different values of $\varepsilon$. Then, we determined the appropriate regularization parameter of the Sinkhorn algorithm, ensuring the Sinkhorn distance is close but no lower than the cost of the solution generated by our algorithm. We recorded the running time and the number of parallel rounds for both Sinkhorn and our algorithm. We also repeated each experiment using a reversed process by fixing the regularization parameter of Sinkhorn and searching for the $\varepsilon$ value for our algorithm, which guarantees the our cost is similar to, but no more than, the Sinkhorn distance; the results for these reversed experimental setups can be found in the technical appendix . Note that, we also recorded the execution time of solving for the exact solution using POT's EMD function, which runs on the CPU. We only present for values of $\varepsilon$ that are large enough such that either our algorithm or Sinkhorn algorithms run faster than the exact algorithm. We also record the additive error, relative to the optimal solution, of both Sinkhorn and our algorithm in appendix section C.1. Additionally, we conducted experiments comparing the sequential performances of our algorithm and Sinkhorn on the CPU, which can be found in the appendix section C.4.

**Synthetic Data:** For synthetic data generation, for both the assignment and OT experiments, we randomly sampled the location of two groups of $n = 10,000$ vertices, $A$ and $B$, in a 2-dimensional unit square. For the assignment problem, the demand or supply of every vertex is $1/n$. For the OT problem, the capacity of each vertex is initially chosen by selecting, uniformly at random from the interval $[0, 1]$. Then, the capacities are normalized such that the total supply and demand are both equal to 1. For any pair of points $(a, b) \in A \times B$, the cost $c(a, b)$ was set as the squared Euclidean distance between them. For each value of $\varepsilon$, we executed 10 runs. For each combination of $\varepsilon$, and algorithm choice, we averaged both the running times as well as the number of parallel rounds over all 10 runs and recorded the results. The results of running times and parallel rounds can be seen in Figure 1(a)(b) and Figure 2(a)(b) respectively.

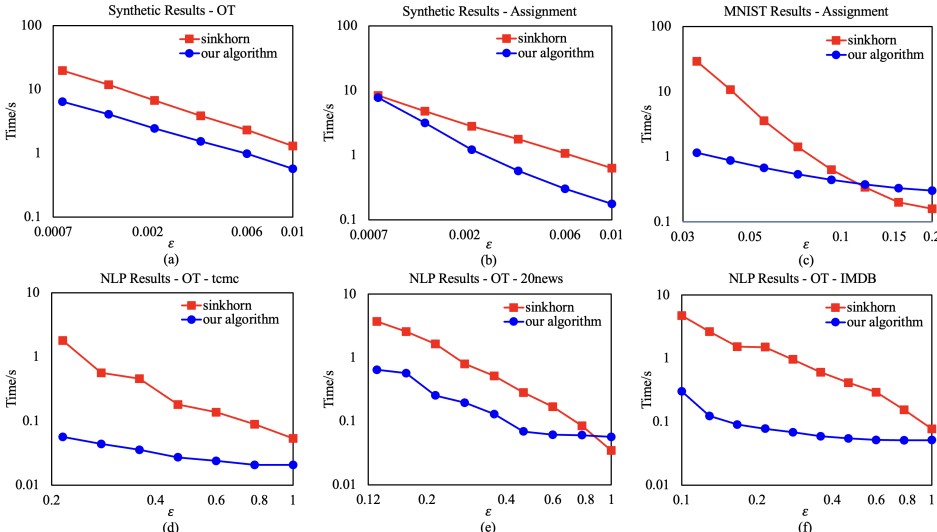

Figure 1: Plots of running times on GPU for the synthetic inputs (a)(b) and the real data inputs (c)(d)(e)(f).

**MNIST:** Next, we ran a similar experiment using real-world image data. We generated our inputs using the MNIST dataset of hand-written digit images [21]. Each image consists of a $28 \times 28$ pixel gray-scale image. The sets $A$ and $B$ each consist of $n = 10,000$ images from the MNIST dataset, selected at random. The cost $c(a, b)$ between two images $a \in A$ and $b \in B$ is computed as follows: Let $a(i, j)$ (resp. $b(i, j)$) be the value of the pixel in row $i$ and column $j$ of image $a$ (resp. $b$). First, the two images are normalized so that the sum of all pixel values is equal to 1 for each image, i.e., $\sum_{i,j \in [1,28]} a(i, j) = 1$ and $\sum_{i,j \in [1,28]} b(i, j) = 1$. Then, the cost $c(a, b)$ is given by the $L_1$ distance between the resulting normalized images: $c(a, b) = \sum_{i,j \in [1,28]} |a(i, j) - b(i, j)|$. Note that an upper

bound on the largest cost is 2. For each algorithm and for each value of $\varepsilon$, we averaged both the running times as well as the number of parallel rounds over 10 runs. The results for these experiments can be found in the plot in Figure 1(c) and the plot in Figure 2(c).

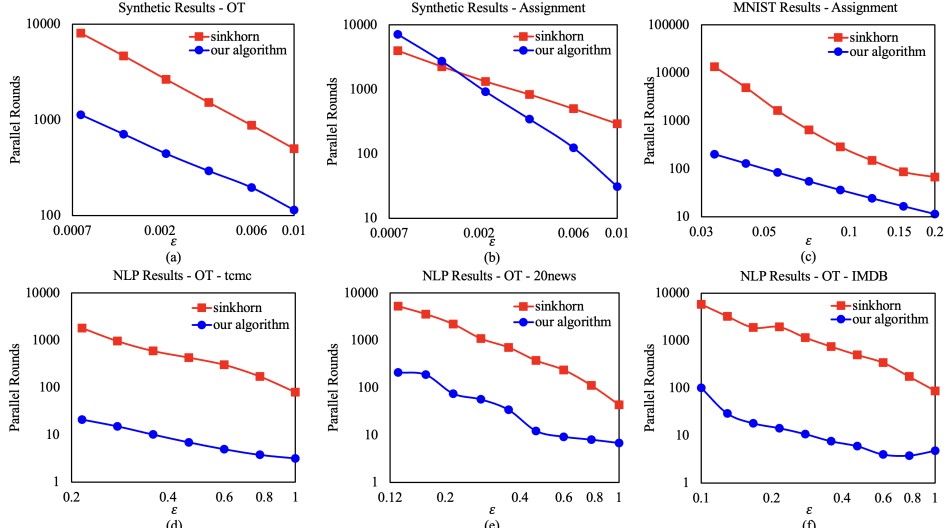

Figure 2: Plots of parallel rounds on GPU for the synthetic inputs (a)(b) and the real data inputs (c)(d)(e)(f).

**NLP:**    In our final experiment, we considered OT with natural language data. We calculate the document distances based on OT following the procedure of previous work [18]. Each OT instance was generated by selecting two discrete sections of text with fixed lengths. Each unique word in the first (resp. second) section of text corresponds to a vertex of $A$ (resp. $B$) in the OT problem. To acquire the supply and demand for each vertex, we tokenized each text section with NLTK [6], and counted the number of appearances of each unique token. Then the counts were normalized so that the total supply and demand were each equal to 1. Next, to generate the costs between vertices, we represent each unique token using a 100-dimensional GloVe word embedding [26]. The cost of any edge $(a, b) \in A \times B$ is then given by the Euclidean distance between the corresponding points in this embedding.

In the last three plots in Figure 1 and Figure 2, we show the results of applying this experimental setup, using sections of the text from *The Count of Monte Cristo*, *20News*, *IMDB*. For each dataset, five different OT instances are created, using different sections of the text, and the results are averaged over all 5 runs. These experiments can be found in Figure 1(d)(e)(f) and Figure 2(d)(e)(f).

In all our experiments, our new parallel combinatorial algorithm almost always runs significantly faster than the Sinkhorn algorithm for the OT problem often with significantly fewer parallel rounds. Unlike the POT's highly optimized implementation of the Sinkhorn method, the implementation of our algorithm is new and may benefit significantly from further optimizations.

## 6 Conclusion

In this work, we provided a fast, highly parallelizable combinatorial algorithm for computing an $\varepsilon$-approximate solution to the assignment problem and the OT problem. We also provided a practical implementation of a slight variation of our algorithm, which outperforms the Sinkhorn algorithm, in terms of running times, in our experimental comparison. In light of this work, we would like to propose the following open question : Is it possible to improve the $O(\log(n)/\varepsilon^2)$ parallel running time of our algorithm, possibly by introducing an appropriate modification? In particular, our simplified practical algorithm presented in Section 4.2 seems to execute fewer parallel rounds than the our worst-case theoretical analysis might suggest. Can the simplifications used in our practical implementation be used to improve our worst-case theoretical running times?

## Acknowledgement

We would like to acknowledge Advanced Research Computing (ARC) at Virginia Tech, which provided us with the computational resources used to run the experiments. Research presented in this paper was funded by NSF CCF-1909171 and NSF CCF-2223871. We would like to thank the anonymous reviewers for their useful feedback.

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
