# OpenReview forum: "A Combinatorial Algorithm for Approximating the Optimal Transport in the Parallel and MPC Settings"
_NeurIPS.cc/2023/Conference — NeurIPS 2023 poster_

### Official Review · Reviewer_i1bb · 2023-06-28

**Soundness:** 3 good
**Presentation:** 3 good
**Contribution:** 3 good
**Rating:** 6
**Confidence:** 4

**Summary:**

This paper gives the first parallel, combinatorial algorithm to approximate the optimal transport distance. Moreover, the algorithm nearly matches the run-time of the best known parallel algorithms for the problem, with expected parallel runtime of O(log(n)/eps^2) (for eps*n the additive error of the OT cost). The motivation for this work comes from the fact that the previous best theoretical guarantee for this problem—an O(log n/eps) runtime parallel algorithm by Jambulapti et al (Neurips 2019)—is not really implementable. Thus practitioners normally use the much easier to implement Sinkhorn algorithm, which has parallel complexity \tilde{O}(log n/eps^O(1)). Additionally, while there is another combinatorial for the sequential problem, it was not able to be parallelized.

Further, in the MPC setting, the authors show that one can  execute their algorithm in the MPC setting with O(log log (n/eps)/ eps^2) rounds with O(n/eps) memory per machine.  Lastly the authors implement their algorithm to exploit GPU parallelism and show that in nearly all settings reported, they beat the Sinkhorn algorithm.

**Strengths:**

- I find the motivation for this paper to be very interesting. While it is in theory important that we push worst-case guarantees (such as the Jambulapti et al work), it is important to design algorithms that are implementable and still have good results. This combinatorial, parallelizeable algorithm certainly does this.
- As someone more combinatorially minded, I liked the algorithm. It’s basically a push-relabel type algorithm.
- Empirically, the proposed algorithm does well enough compared to the Sinkhorn algorithm that it might even be that the analysis in the paper is a bit lossy. Empirical evidence was convincing enough for me.
- Well Written

**Weaknesses:**

- In terms of raw numbers, there’s no big theoretical improvement. (But again, that’s not the main point of the paper.)
- The presentation in the paper could be better at times. To be clear, the writing is very very good, but the algorithm is so simple in a nice way, that I think they could’ve made points clearer. I will elaborate in comments below.

**Questions:**

Comments to the authors:
- Remove sentence in line 21-23, you already say this at a more relevant part later
- R_{>0} -> \mathb{R}_{>0} in line 30
- Line 72 “achieve a improved” -> “achieve an improved”
- Is the Sinkhorn runtime tight around O(log n/eps^O(1))? Or empirically is it better than that guarantee? Also can you explicitly state whatever constant the O(1) term is hiding at least once?
- Line 179 “which will has a ” -> “which has a ”
- This algorithm is so reminiscent of push-relabel that I kept looking for more explicit direct comparisons to push-relabel in section 2. In fact step 2 of the algorithm is just an augmenting path and you keep an eps-feasible dual, similar to the heights in the push-relabel algorithm. Then even your proof of correctness is the same as push-relabel. Can you say more about the inspiration from push-relabel?
- You should add a figure for the algorithm, maybe 3 sub-figures, one for each of the 3 steps
- Line 308 “we have describe our” ->  “we have described our”
- I think the lines you refer to in algorithm 1 are all messed up. Double check me on this but I think where you say “Lines 5-9” should be “Lines 5-7”, where you say “Lines 10-13” should be “Lines 8-10”, and where you say “Lines 14-20” should be “Lines 11-15”.
- Lime 694 in appendix needs to be rewritten

---

> ### Author Rebuttal · Authors · 2023-08-10
>
>
> *Is the Sinkhorn runtime tight around $O(\log n/\varepsilon^{O(1)})$? ... Also can you explicitly state whatever constant the $O(1)$ term is hiding at least once?* Over time, the analysis of the Sinkhorn algorithm has seen multiple improved bounds to its $\varepsilon$ dependency. To our knowledge, the best bound is given by Dvurechensky et al. (see below for citation), which gives a sequential $\tilde{O}(n^2/\varepsilon^2)$ bound, and is easily parallelizable to $\tilde{O}(1/\varepsilon^2)$, basically matching our algorithm's provable bounds. (We use $\tilde{O}$ to hide logarithms here). We will edit the paper to clarify this.
>
> *Or empirically is it (Sinkhorn) better than that guarantee?* This seems to be the case, based on our results, and prior work with Sinkhorn that we have seen. Our algorithm seems to also outperform its own theoretical expectations.
>
> *This algorithm is so reminiscent of push-relabel that ...*
> Thank you for pointing this out. Indeed, our algorithm is using the popular push-relabel framework and our proof of correctness is also based on the standard feasibility conditions for matchings. Our main technical novelty is in the efficiency analysis of our algorithm. Using the fact that edge costs are small integer multiples of $\varepsilon$, we bound the number of push-relabel iterations by $1/\varepsilon^2$. This allows us to achieve a parallel complexity of $O(\log n/\varepsilon^2)$. We will include this in the discussion of "Our Approach" (L131-141).
>
> Dvurechensky, P., Gasnikov, A., Kroshnin, A. (2018, July). Computational optimal transport: Complexity by accelerated gradient descent is better than by Sinkhorn’s algorithm. In International conference on machine learning (pp. 1367-1376). PMLR.

---

> > ### Comment · Reviewer_i1bb · 2023-08-15
> >
> > Thank you for the response. My score remains the same for now, but I will continue to monitor the discussion during this response period.

---

### Official Review · Reviewer_dyx3 · 2023-07-03

**Soundness:** 4 excellent
**Presentation:** 4 excellent
**Contribution:** 3 good
**Rating:** 6
**Confidence:** 3

**Summary:**

This paper presents a novel combinatorial algorithm that finds an $\epsilon$-optimal transport plan for the optimal transport problem. Additionally, it introduces a variant of this algorithm in the MPC model, exhibiting an expected $O(\log\log n)$ communication rounds, with each machine having a memory of $O(n)$ order. This algorithm offers speedups over previous approaches, which required at least $\Omega(\log n)$ rounds, in the large-$n$ scenarios. Furthermore, the authors provided a GPU-friendly interpretation of their algorithm utilizing matrix operations, and supported their results using numerical experiments.

**Strengths:**

The primary contribution of this paper is a combinatorial algorithm for approximate optimal transport, which exhibits efficient parallelization and significantly outperforms previous algorithms in the Massive Parallel Computation (MPC) model with respect to $n$, the total support size of the input distributions. The algorithm is simple and elegant, allowing for a GPU-friendly interpretation based on matrix operations, thereby enhancing its competitiveness in real-world scenarios.

The analysis of the algorithm is relatively straightforward and easy to follow, while the conducted numerical experiments provide compelling evidence for its effectiveness.

**Weaknesses:**

1. This paper only improves upon previous works in the MPC model rather than the broader parallel settings. In the context of the parallel time, both this work and previous algorithms based on Sinkhorn-Knopp methods are of order $O(\log n)$. Consequently, this limitation might restrict the applicability of this algorithm in real-world scenarios with different parallelization settings.

2. Compared to previous works, this algorithm has a larger dependence on $\epsilon$.

**Questions:**

Correspondingly, I have two questions that, if addressed, could make the results even stronger:

1. Using similar combinatoric techniques, is it possible to obtain an algorithm that has better performance than previous works in parallel time?
2. Can further improvements be made to reduce the dependence on $\epsilon$, or are there practical reasons why the focus remains primarily on the dependence on $n$?

Minor comments:
1. It would be good to have a discussion on open questions.
2. Page 2, line 40: "one" -> "when"
3. Please check the format of reference [16,22,23,24]. In particular, the title of [16] and the author lists of [22-24] are not presented correctly.

**Limitations:**

Not relevant in my opinion.

---

> ### Author Rebuttal · Authors · 2023-08-10
>
>
> *Using similar combinatoric techniques, is it possible to obtain an algorithm that has better performance than previous works in parallel time?*
>
> We believe the improving our parallel OT epsilon dependency from $O(1/\varepsilon^2)$ to $O(1/\varepsilon)$ is plausible, but difficult – we have spent considerable time trying to do so. Designing such an algorithm remains an interesting open question.
>
> *Can further improvements be made to reduce the dependence on $\varepsilon$, or are there practical reasons why the focus remains primarily on the dependence on $n$?*
>
> Typically, in many applications, one may be happy with obtaining OT-cost with a precision of two decimal points, i.e., $\varepsilon = 0.01$. In contrast, the input size may vary by large numbers. For instance, an image with $n$ pixels can be treated as a distribution with $n$ points in its support. The number of pixels in different images may vary by a lot. Therefore, typically, it is desirable to have algorithms with better dependence on the input size as opposed to a better dependence on the error parameter.
>
> *It would be good to have a discussion on open questions.* Yes, we can add one using the extra page of space in the final version.

---

> > ### Comment · Reviewer_dyx3 · 2023-08-14
> >
> > I would like to thank the authors for the detailed rebuttal. I remain my rating.

---

### Official Review · Reviewer_pKrQ · 2023-07-06

**Soundness:** 4 excellent
**Presentation:** 4 excellent
**Contribution:** 2 fair
**Rating:** 6
**Confidence:** 2

**Summary:**

The paper presents a parallel combinatorial algorithm for the optimal transport (OT) and minimum weight perfect matching. The algorithm computes an additive epsilon approximation.

The algorithm runs in O(n^2 / epsilon^O(1)) time and (in a straightforward way) gives an algorithm running in O(log log n / epsilon^O(1)) MPC rounds, using linear space per machine.

The paper also compares a GPU implementation of the algorithm with the Sinkhorn method showing improved running time.

**Strengths:**

* The paper gives a simple and clean algorithm for a practically relevant problem.
* The algorithm delivers good speedups over the baseline

**Weaknesses:**

The main weakness of the paper is a very limited empirical evaluation. The 1.5-page long section on empirical evaluation contains mostly the dataset descriptions (which in my opinion could easily go to the supplementary material if space is a concern) + a short running time comparison to a single baseline. I'd like to see a more in-depth comparison, for example (I'm trying to suggest ideas here, not define a list of requirements):
* a comparison between CPU implementations
* analysis of the number of rounds used
* evaluation on non-dense graphs
* quality analysis: what is the actual absolute / relative error as a function of epsilon

Minor suggestions:
 * I don't think describing the Israeli-Itai algorithm is a good use of the space (I'd suggest removing it), given that the algorithm ends up using a different idea (which I consider folklore)

**Questions:**

1. What is the work and depth (aka span) of the algorithm described in 4.2? (in the shared-memory work-depth model)
1. What is the reason for not including DROT results in section 5? How does DROT compare to Sinkhorn?

**Limitations:**

Yes

---

> ### Author Rebuttal · Authors · 2023-08-10
>
>
> *The paper presents a parallel combinatorial algorithm for the optimal transport (OT) and maximum weight matching.*
> We believe that the reviewer has misunderstood the problem being solved in this paper.
> We present an algorithm for the *minimum* weight matching (i.e., the assignment problem) which is a significantly harder to approximate than the *maximum* weight matching. For instance, the greedy algorithm is known to provide a $1/2$ approximation for the maximum weight matching problem. However, for minimum weight matching, the greedy approach has an unbounded approximation ratio and, even for points on a line, the approximation ratio improves only to $\Omega(n^{0.59})$ (see citation below). Given the reviewer's misunderstanding of the problem, we believe they might have given an inaccurate evaluation of the paper.
>
> *It is possible to black-box reduce maximal matching to $(1+\varepsilon)$ approximate matchings in MPC (see Corollary 1.3 of ...* Again, this result is for approximating the maximum weighted matching, which, as we discussed previously is a different problem than minimum-weight matching.
>
> *I'd like to see a more in-depth comparison, for example ... CPU implementations.*
> We have already provided a CPU comparison results for our sequential algorithm; see appendix section C.3.
>
> *Analysis of the number of rounds used ...*
> This is a good suggestion, one that we should have included, because it gives an implementation-independent gauge of algorithm convergence. We reran all of our experiments. For OT, we found that our algorithm consistently executes something like 10x fewer iterations than Sinkhorn. For assignment problem, it is more mixed. This corresponds to what we observe in our actual running times, and demonstrates that our speedup is intrinsic, rather than simply a matter of implementation. For the final version of the paper, we will add iteration counts for all experiments.
>
> *Evaluation on non-dense graphs ...*
> The OT and assignment problems have been formulated on complete graphs. So, testing the algorithm on sparse graphs is not applicable for the problem.
>
> *Quality analysis ...* In each of our experiments, we already do compare the actual errors produced by the algorithms involved, not just the input values of the error parameters. In the Sinkhorn comparisons, we select the error parameter input value so that the actual error produced by Sinkhorn is within 1E-6 of our algorithm’s actual error. Thus, the current comparison is fair in terms of the actual produced solution quality. However, we understand that including concrete values for the actual errors produced in comparison to the optimal may still be valuable information, and we will update the figures to include this information.
>
> *What is the work and depth (aka span) of the algorithm described in 4.2?*
>
> As mentioned in Section 4.1 L262--266, plugging in the parallel algorithm by Israeli-Itai to our algorithm from Section 3 leads to a parallel implementation with a depth of $O(\log n/\varepsilon^2)$. The work is dependent on the total work done by all invocations of the Itai Israeli algorithm. While they do not specifically give a bound on work, they do assume at most $|E|$ processors are used, so the work is $|E| \log |E|$, where $E$ is the set of edges in the admissible graph. Using equation (4), the total work can therefore be bounded by $O(n^2 \log(n) / \varepsilon)$.
>
> The modifications presented in Section 4.2 are motivated by a need for practical implementation on the GPU. In particular, we adapt and simplify the Israeli-Itai's algorithm for general graphs to the bipartite setting. Although, we haven't formally analyzed the span of this practical implementation, we believe it will be the same as that of the standard implementation. After a thorough check, we will include a discussion on the span of this implementation in the next version of the paper.
>
> *What is the reason for not including DROT results in section 5? How does
> DROT compare to Sinkhorn?*
> We have included a comparison of DROT with our algorithm in appendix section C.2 and our algorithm generally outperforms DROT.
>
> The implementation of DROT is in CUDA whereas our algorithm as well as Sinkhorn are implemented in PyTorch. CUDA implementations are faster than those on PyTorch, and despite this, our algorithm generally outperformed DROT. Nonetheless, due to a difference in implementation platform, we decided not include these results in Section 5 but to move it to the appendix. For the same reason, we decided not to compare DROT against Sinkhorn.
>
> Citation :
> Reingold, E. M., Tarjan, R. E. (1981). On a greedy heuristic for complete matching. SIAM Journal on Computing, 10(4), 676-681.

---

> > ### Comment · Reviewer_pKrQ · 2023-08-14
> > **Thank you for the feedback**
> >
> > Thank you for the response. I'm sorry for confusing the maximum with minimum weight variants, thank you for pointing this out.
> >
> > I think the paper should be accepted and will update the review accordingly.

---

### Official Review · Reviewer_mqPb · 2023-07-15

**Soundness:** 3 good
**Presentation:** 3 good
**Contribution:** 3 good
**Rating:** 6
**Confidence:** 3

**Summary:**

The paper considers the Optimal Transport problem, which is a metric for the distance between distributions, used by some ML/DL methods. It presents a combinatorial algorithm for that problem, which can be parallelized and implemented efficiently both in map-reduce systems (MPC) and on a GPU. It also addresses a special case of that problem, the assignment problem.
The paper proves that the worst-case run-time complexity of the presented algorithm is O(n^2/ \epsilon^2) (for assignment O(n^2/ \epsilon)) in sequential time. The parallel time is O(\log n / \epsilon ^2). This improves upon the Sinkhorn-Knopp method, which is the state-pf-the-art method for this problem used in the Python optimal transport package. Although this run-time does not improve upon the theoretic results of Jambulapati et al., the authors explain that their algorithm is difficult to implement and not used. For MPC the paper shows that O(\loglog n / \epsilon ^2) rounds are required with O(n) memory per machine, the first sub-logarithmic combinatorial algorithm for that problem.
Finally, the paper presents experimental results on a GPU, showing significant run-time improvements (sometimes above 10x) over the Sinkhorn-Knopp method for six datasets in most of the relevant ranges of epsilon (two for assignment, four for optimal transport).

**Strengths:**

The paper considers an interesting problem and clearly presents a new algorithm for solving it, showing both theoretic and experimental results. The new results improve upon the previous methods published in NeurIPS several years ago. The algorithm creatively combines and modifies existing ideas. The presented results are comprehensive, addressing different systems on which the algorithm may run. The special case of the assignment problem is interesting on its own.

**Weaknesses:**

The paper has several weaknesses:
- The code used for the experiments was not provided, so the results cannot be reproduced and might not benefit the community.
- The algorithm of Jambulapati, which is the best asymptotically based on theoretical analysis (for parallel time), was not a part of the experimental comparison.
- The experimental results use "toy examples" of datasets with some ranges of epsilon. It is unclear if the presented algorithm is the best choice for practical scenarios, such as those in the cited papers that use optimal transport (or whether linear programming or the exact algorithm could be the best choice there).
- Although this paper improves upon papers published in NeurIPS several years ago, it may have a bigger relevant audience in conferences focused on combinatorial algorithms.



**Questions:**

Questions to the authors:
- Can you share the code, to enable reproducibility and benefit the community?
- Can you present an experimental comparison with the algorithm of Jambulapati et al.?
- Can you present an experimental comparison in one of the real scenarios in which optimal transport was used in ML/DL? (e.g., from one of the papers cited)

**Limitations:**

Yes, I don't see an issue regarding this.

---

> ### Author Rebuttal · Authors · 2023-08-10
>
>
> *The code used for the experiments was not provided...* Actually, our code was provided as part of the supplement. You can find it in the provided zip file, alongside the appendix PDF. The code also includes a detailed README that explains how to run the experiments, specifically meant to facilitate reproducibility.
>
> *The algorithm of Jambulapati ... was not a part of the experimental comparison.* Although the work by Jambulapati et al. gives the asymptotically fastest bound in theory, there is good reason to believe that the overhead of the algorithm causes it to be less efficient than Sinkhorn in practice. Lin et al. [21] agree with this assessment, stating: "Despite the theoretically sound complexity bound, the lack of simplicity and ease-of implementation make this algorithm less competitive with Sinkhorn and Greenkhorn which remain the baseline solution methods in practice.'' Even the Jambulapati et al. paper itself does not claim to outperform Sinkhorn.  Overall, since Sinkhorn remains a better baseline in practice, we did not see a need to compare to the Jambulapati et al. Furthermore, we are not aware of any publicly available GPU implementation of their algorithm.
>
> *Can you present an experimental comparison in one of the real scenarios in which optimal transport was used in ML/DL?* This was the intent of our current NLP experiments, which are similar in principle to the so-called `Word Movers Distance', a very well known method of comparing documents. See the paper ``From Word Embeddings To Document Distances'', which appeared in the 32nd International Conference on Machine Learning (ICML) and has over 2000 citations.

---

> > ### Comment · Reviewer_mqPb · 2023-08-18
> > **Thank you**
> >
> > Thank you for your response, which indeed addresses my questions and comments. In view of concerns raised by other reviewers I currently keep my rating, but I will follow the discussion and consider raising it.

---

### Author Rebuttal · Authors · 2023-08-10

We would like to thank all the reviewers for their helpful suggestions. The reviewers pointed out some minor grammatical and presentation changes; we appreciate these suggestions and will address them in the final paper version. For questions and comments that warrant a more significant discussion, we have addressed our responses directly to the relevant reviewer. When addressing such questions or comments, we will quote directly from the reviewer, using italics, in order to concisely reference the relevant portion of the review.

---

### Decision · Program_Chairs · 2023-09-21

**Decision:**

Accept (poster)

**Comment:**

The paper provides an elegant parallel algorithm that is easy to implement and it outperforms the popular Sinkhorn algorithm in the experiments. Compared to the state of the art theoretical algorithm, the proposed algorithm is more practical and easier to implement. The reviewers appreciated the algorithm's simplicity and practical merit. Overall, the paper makes a valuable contribution to this line of work that is beneficial to the community.